# Closure Discovery for Coarse-Grained Partial Differential Equations Using Grid-based Reinforcement Learning

Jan-Philipp von Bassewitz[1,2]    Sebastian Kaltenbach[1,2]    Petros Koumoutsakos[2] *
[1]ETH Zurich    [2]Harvard SEAS

Reliable predictions of critical phenomena, such as weather, wildfires and epidemics often rely on models described by Partial Differential Equations (PDEs). However, simulations that capture the full range of spatio-temporal scales described by such PDEs are often prohibitively expensive. Consequently, coarse-grained simulations are usually deployed that adopt various heuristics and empirical closure terms to account for the missing information. We propose a novel and systematic approach for identifying closures in under-resolved PDEs using grid-based Reinforcement Learning. This formulation incorporates inductive bias and exploits locality by deploying a central policy represented efficiently by a Fully Convolutional Network (FCN). We demonstrate the capabilities and limitations of our framework through numerical solutions of the advection equation and the Burgers' equation. Our results show accurate predictions for in- and out-of-distribution test cases as well as a significant speedup compared to resolving all scales.

## 1. Introduction

Simulations of critical phenomena such as climate, ocean dynamics and epidemics, have become essential for decision-making, and their veracity, reliability, and energy demands have great impact on our society. Many of these simulations are based on models described by PDEs expressing system dynamics that span multiple spatio-temporal scales. Examples include turbulence [1], neuroscience [2], climate [3] and ocean dynamics [4].

Today, we benefit from decades of remarkable efforts in the development of numerical methods, algorithms, software, and hardware and witness simulation frontiers that were unimaginable even a few years ago [5]. Large-scale simulations that predict the system's dynamics may use trillions of computational elements [6] to resolve all spatio-temporal scales, but these often address only idealized systems and their computational cost prevents experimentation and uncertainty quantification.

By contrast, reduced order and coarse-grained models are fast, but limited by the linearization of complex system dynamics while their associated closures, which model the effect of unresolved (not simulated) dynamics on the quantities of interest, are in general based on heuristics and, as a result, domain specific [7]. A closure discovery framework that is independent of the system of interest and can be applied to various domains and tasks is thus highly desirable [8].

To address this challenge we propose *Closure-RL*, a framework to complement coarse-grained simulations with closures that are discovered by a grid-based Reinforcement Learning (RL) framework. Closure-RL employs a central policy that is based on a FCN and uses locality as an inductive bias. It is able to correct the error of the numerical discretization and improves the overall accuracy of the coarse-grained simulation.

Our approach is inspired by pixelRL [9], a recent work in Reinforcement Learning (RL) for image reconstruction, which minimizes local reconstruction errors. PixelRL employs a *per-pixel* reward and value network and can therefore be interpreted as a cooperative Multi-Agent Reinforcement Learning (MARL) framework with one agent per pixel. We extend this framework to closure discovery by

---

*Corresponding Author: `petros@seas.harvard.edu`

Second Conference on Parsimony and Learning (CPAL 2025).

treating the latter as a reconstruction problem. The numerical scheme introduces corruptions, that the agents are learning to reverse. In contrast to actions based on a set of filters as in pixelRL, we employ a continuous action space that is independent of the PDE to be solved. Our agents learn and act locally on the grid, in a manner that is reminiscent of the numerical discretizations of PDEs based on Taylor series approximations. Hereby, each agent only gets information from its spatial neighborhood.

Although Closure-RL has many characteristics of a cooperative MARL approach, the training costs are similar to a single-agent RL formulation as our proposed formulation employs a central policy and does not require complex interactions between the agents. This allows us to use a large number of agents and no fine-tuning regarding the placement of the agents is required. After training, the framework is capable of accurate predictions in both in- and out-of-distribution test cases. We remark that the actions taken by the agents are highly correlated with the numerical errors and can be viewed as an implicit correction to the effective convolution performed via the coarse-grained discretization of the PDEs [10]. We note that we have chosen a RL-based approach to not require access to a differentiable solver or potentially difficult gradient computations via the adjoint method [8]. Our approach is non-intrusive and can be seamlessly integrated with any numerical discretization scheme.

The main contribution of this work is a grid-based RL algorithm for the discovery of closures for coarse-grained PDEs. The algorithm provides an automated process for the correction of the errors of the associated numerical discretization. We show that the proposed method is able to capture scales beyond the training regime and provides a potent method for solving PDEs with high accuracy and limited resolution.

## 2. Related Work

**Machine Learning and Partial Differential Equations:** In recent years, there has been significant interest in learning the solution of PDEs using Neural Networks. Techniques such as PINNs [11, 12], DeepONet [13], the Fourier Neural Operator [14], NOMAD [15], Clifford Neural Layers [16] and an invertible formulation [17] have shown promising results for both forward and inverse problems. However, there are concerns about their accuracy and related computational cost, especially for low-dimensional problems [18]. These methods aim to substitute numerical discretizations with neural nets, in contrast to our RL framework, which aims to complement them. Moreover, their loss function is required to be differentiable, which is not necessary for the stochastic formulation of the RL reward.

**Reinforcement Learning:** The present approach is designed to solve various PDEs using a central policy and it is related to similar work for image reconstruction [9]. Its efficient grid-based formulation is in sharp contrast to multi-agent learning formulations that train agents on decoupled subproblems or learn their interactions [19–23]. The single focus on the local discretization error allows for a general method that is not required to be fine-tuned for the actual application by selecting appropriate global data (such as the energy spectrum) [24] or domain-specific agent placement [25].

**Closure Modeling:** The development of machine learning methods for discovering closures for under-resolved PDEs has gained attention in recent years. Current approaches are mostly tailored to specific applications such as turbulence modeling [24–27] and use data such as energy spectra and drag coefficients of the flow in order to train the RL policies. In [28], a more general framework based on diffusion models is used to improve the solutions of Neural Operators for temporal problems using a multi-step refinement process. Their training is based on supervised learning in contrast to the present RL approach which additionally complements existing numerical methods instead of neural network based surrogates [14, 29].

**Inductive Bias:** The incorporation of prior knowledge regarding the physical processes described by the PDEs, into machine learning algorithms is critical for their training in the Small Data regime and for increasing the accuracy during extrapolative predictions [30]. One way to achieve this is by shaping appropriately the loss function [12, 31–34], or by incorporating parameterized mappings that are based on known constraints [35–37]. Our RL framework incorporates locality and is thus consistent with numerical discretizations that rely on local Taylor series based approximations. The

incorporation of inductive bias in RL has also been attempted by focusing on the beginning of the training phase [38, 39] in order to shorten the exploration phase.

# 3. Methodology

We consider a time-dependent, parametric, non-linear PDE defined on a regular domain $\Omega$. The solution of the PDE depends on its initial conditions (ICs) as well as its PDE-parameters (PDEP) $C \in \mathcal{C}$. The PDE is discretized on a spatiotemporal grid and the resulting discrete set of equations is solved using numerical methods.

In turn, the number of the deployed computational elements and the structure of the PDE determine whether all of its scales have been resolved or whether the discretization amounts to a coarse-grained representation of the PDE. In the first case, the **fine grid simulation (FGS)** provides the discretized solution $\psi$, whereas in **coarse grid simulation (CGS)** the resulting approximation is denoted by $\tilde{\psi}$.[2] The RL policy can improve the accuracy of the solution $\tilde{\psi}$ by introducing an appropriate forcing term in the right-hand side of the CGS. For this purpose, FGS of the PDE are used as training episodes and serve as the ground truth to facilitate a reward signal. The CGS and FGS employed herein are introduced in the next section. The proposed grid-based RL framework is introduced in Section 3.2.

## 3.1. Coarse and Fine Grid Simulation

We consider a FGS of a spatiotemporal PDE on a Cartesian 3D grid with temporal resolution $\Delta t$ for $N_T$ temporal steps with spatial resolution $\Delta x, \Delta y, \Delta z$ resulting in $N_x, N_y, N_z$ discretization points. The CGS entails a coarser spatial discretization $\widetilde{\Delta x} = d\Delta x$, $\widetilde{\Delta y} = d\Delta y$, $\widetilde{\Delta z} = d\Delta z$ as well as a coarser temporal discretization $\widetilde{\Delta t} = d_t \Delta t$. Here, $d$ is the spatial and $d_t$ the temporal scaling factor. Consequently, at a time-step $\tilde{n}$, we define the discretized solution function of the CGS as $\tilde{\psi}^{\tilde{n}} \in \tilde{\Psi} := \mathbb{R}^{k \times \tilde{N}_x \times \tilde{N}_y \times \tilde{N}_z}$ with $k$ being the number of solution variables. The corresponding solution function of the FGS at time-step $n_f$ is $\psi^{n_f} \in \Psi := \mathbb{R}^{k \times N_x \times N_y \times N_z}$. The discretized solution function of the CGS can thus be described as a subsampled version of the FGS solution function and the subsampling operator $\mathcal{S} : \Psi \to \tilde{\Psi}$ connects the two.

The time stepping operator of the CGS $\mathcal{G} : \tilde{\Psi} \times \tilde{\mathcal{C}} \to \tilde{\Psi}$ leads to the update rule

$$\tilde{\psi}^{\tilde{n}+1} = \mathcal{G}(\tilde{\psi}^{\tilde{n}}, \tilde{C}^{\tilde{n}}). \tag{1}$$

Similarly, we define the time stepping operator of the FGS as $\mathcal{F} : \Psi \times \mathcal{C} \to \Psi$. In the case of the CGS, $\tilde{C} \in \tilde{\mathcal{C}}$, is an adapted version of $C$, which for instance involves evaluating a PDE-parameter function using the coarse grid.

To simplify the notation, we set $n := \tilde{n} = d_t n_f$ in the following. The update rule of the FGS, i.e. applying $\mathcal{F}$ $d_T$-times to advance for the same time increment then the CGS, is referred to as

$$\psi^{n+1} = \mathcal{F}^{d_t}(\psi^n, C^n). \tag{2}$$

In Section 4.1 and Section 4.2 we are introducing the FGS and CGS for our illustrative examples. In line with our experiments and to further simplify notation, we are dropping the third spatial dimension in the following presentation.

## 3.2. RL Environment

The environment of our RL framework is summarized in Figure 1. We define the state at step $n$ of the RL environment as the tuple $S^n := (\psi^n, \tilde{\psi}^n)$. This state is only partially observable as the policy is acting only in the CGS. The observation $O^n := (\tilde{\psi}^n, \tilde{C}^n) \in \mathcal{O}$ is defined as the coarse representation of the discretized solution function and the PDEPs. Our goal is to train a policy $\pi$ that makes the dynamics of the CGS to be close to the dynamics of the FGS. To achieve this goal, the action $A^n \in \mathcal{A} := \mathbb{R}^{k \times \tilde{N}_x \times \tilde{N}_y}$ at step $n$ of the environment is a collection of forcing terms for each

---

[2]In the following, all variables with the ˜ are referring to the coarse-grid description.

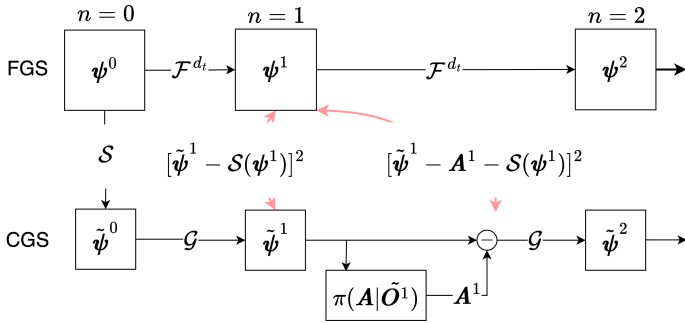

Figure 1: Illustration of the training environment with the agents embedded in the CGS. The reward measures how much the action taken improves the CGS.

discretization point of the CGS. In case the policy is later used to complement the CGS simulation the update function in Equation (1) changes to

$$\tilde{\psi}^{n+1} = \mathcal{G}(\tilde{\psi}^n - A^n, \tilde{C}^n). \tag{3}$$

To encourage learning a policy that represents the non-resolved spatio-temporal scales, the reward is based on the difference between the CGS and FGS at time step $n$. In more detail, we define a local reward $R^n \in \mathbb{R}^{\tilde{N}_x \times \tilde{N}_y}$ inspired by the reward proposed for image reconstruction in [9]:

$$R^n_{ij} = \frac{1}{k} \sum_{w=1}^{k} \left( [\tilde{\psi}^n - \mathcal{S}(\psi^n)]^2 - [\tilde{\psi}^n - A^n - \mathcal{S}(\psi^n)]^2 \right)_{wij} \tag{4}$$

Here, the square $[\cdot]^2$ is computed per matrix entry. We note that the reward function therefore returns a matrix that gives a scalar reward for each discretization point of the CGS. If the action $A^n$ is bringing the discretized solution function of the CGS $\tilde{\psi}^n$ closer to the subsampled discretized solution function of the FGS $\mathcal{S}(\psi^n)$, the reward is positive, and vice versa. In case $A^n$ corresponds to the zero matrix, the reward is the zero matrix as well.

During the training process, the objective is to find the optimal policy $\pi^*$ that maximizes the mean expected reward per discretization point given the discount rate $\gamma$ and the observations:

$$\pi^* = \underset{\pi}{\operatorname{argmax}} \, \mathbb{E}_\pi \left( \sum_{n=0}^{\infty} \gamma^n \bar{r}^n \right) \tag{5}$$

with the mean expected reward

$$\bar{r}^n = \frac{1}{\tilde{N}_x \cdot \tilde{N}_y} \sum_{i,j=1}^{\tilde{N}_x, \tilde{N}_y} R^n_{ij}. \tag{6}$$

### 3.3. Grid-based RL Formulation

The policy $\pi$ predicts a local action $A^n_{i,j} \in \mathbb{R}^k$ at each discretization point which implies a very high dimensional continuous action space. Hence, formulating the closures with a single agent is very challenging. However, since the rewards are designed to be inherently local, locality can be used as inductive bias and the RL learning framework can be interpreted as a multi-agent problem [9]. One agent is placed at each discretization point of the coarse grid with a corresponding local reward $R^n_{ij}$. We remark that this approach augments adaptivity as one can place extra agents at additional, suitably selected, discretization points.

Each agent develops its own optimal policy, which we later defined to be shared, and Equation (5) is replaced by

$$\pi^*_{ij} = \underset{\pi_{ij}}{\operatorname{argmax}} \, \mathbb{E}_{\pi_{ij}} \left( \sum_{n=0}^{\infty} \gamma^n R^n_{ij} \right), \tag{7}$$

Here, we used $\mathcal{O}(\tilde{N}_x\tilde{N}_y)$ agents, which for typically used grid sizes of numerical simulations, becomes a large number compared to typical MARL problem settings [22].

We parametrize the local policies using neural networks. However, since training this many individual neural nets can become computationally expensive, we parametrize all the agents together using one fully convolutional network (FCN) [40], which implies that the agents share one policy.

### 3.4. Parallelizing Local Policies with a FCN

All local policies are parametrized using one FCN such that one forward pass through the FCN computes the forward pass for all the agents at once. This approach enforces the aforementioned locality and the receptive field of the FCN corresponds to the spatial neighborhood that an agent at a given discretization point can observe.

We define the collection of policies in the FCN as $\Pi^{FCN} : \mathcal{O} \to \mathcal{A}$. In further discussions, we will refer to $\Pi^{FCN}$ as the *global policy*. The policy $\pi_{ij}$ of the agent at point $(i,j)$ is subsequently implicitly defined through the global policy as $\pi_{ij}(O_{ij}) := \left[\Pi^{FCN}(\boldsymbol{O})\right]_{:ij}$. Here, $\boldsymbol{O}_{ij}$ contains only a part of the input contained in $\boldsymbol{O}$ [3]. For consistency, we will refer to $\boldsymbol{O}_{ij}$ as a *local observation* and $\boldsymbol{O}$ as the *global observation*. We note that the policies $\pi_{ij}(\boldsymbol{O}_{ij})$ map the observation to a probability distribution at each discretization point (see Appendix A for details including the neural network architecure employed). Similar to the global policy, the global value function is parametrized using a FCN as well. It maps the global observation to an expected return $H \in \mathcal{H} := R^{\tilde{N}_x \times \tilde{N}_y}$ at each discretization point $\mathcal{V}^{FCN} : \mathcal{O} \to \mathcal{H}$. Similarly to the local policies, we define the *local value function* related to the agent at point $(i,j)$ as $v_{ij}(O_{ij}) := [\mathcal{V}^{FCN}(\boldsymbol{O})]_{ij}$.

### 3.5. Policy Optimization

In order to solve the optimization problem in Equation (7), we employ a modified version of the PPO algorithm [41].

Policy updates are performed by taking gradient steps on

$$\mathbb{E}_{\boldsymbol{S},\boldsymbol{A}\sim\Pi^{FCN}}\left[\frac{1}{\tilde{N}_x \cdot \tilde{N}_y}\sum_{i,j=1}^{\tilde{N}_x,\tilde{N}_y} L_{\pi_{ij}}(\boldsymbol{O}_{ij},\boldsymbol{A}_{ij})\right] \tag{8}$$

with the local version of the PPO objective $L_{\pi_{ij}}(\boldsymbol{O}_{ij},\boldsymbol{A}_{ij})$. This corresponds to the local objective of the policy of the agent at point $(i,j)$ [41].

The global value function is trained on the MSE loss

$$L_{\mathcal{V}}(\boldsymbol{O}^n,\boldsymbol{G}^n) = ||\mathcal{V}^{FCN}(\boldsymbol{O}^n) - \boldsymbol{G}^n||_2^2$$

where $\boldsymbol{G}^n \in \mathbb{R}^{\tilde{N}_x \times \tilde{N}_y}$ represents the actual global return observed by interaction with the environment and is computed as $\boldsymbol{G}^n = \sum_{i=n}^{N} \gamma^{i-n}\boldsymbol{R}^i$. Here, $N$ represents the length of the respective trajectory. We have provided an overview of the modified PPO algorithm in Appendix B together with further details regarding the local version of the PPO objective $L_{\pi_{ij}}(\boldsymbol{O}_{ij},\boldsymbol{A}_{ij})$. Our implementation of the adapted PPO algorithm is based on the single agent PPO algorithm of the Tianshou framework [42].

### 3.6. Computational Complexity

We note that the computational complexity of the CGS w.r.t. the number of discretization points scales with $\mathcal{O}(\tilde{N}_x\tilde{N}_y)$. As one forward pass through the FCN also scales with $\mathcal{O}(\tilde{N}_x\tilde{N}_y)$ the same is true for Closure-RL. The FGS employs a finer grid, which leads to a computational cost that scales with $\mathcal{O}(d_t d^2 \tilde{N}_x\tilde{N}_y)$. This indicates a scaling advantage for Closure-RL compared to a FGS. Based on these considerations and as shown in the experiments, Closure-RL is able to compress some of the computations that are performed on the fine grid as it is able to significantly improve the CGS while keeping the execution time below that of the FGS.

---

[3]The exact content of $\boldsymbol{O}_{ij}$ is depending on the receptive field of the FCN

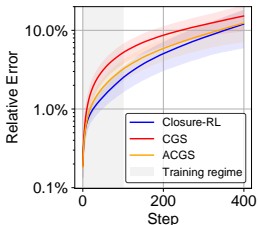 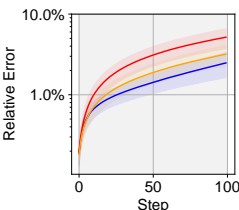 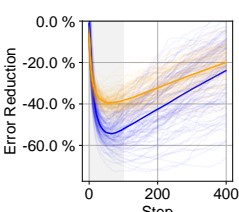 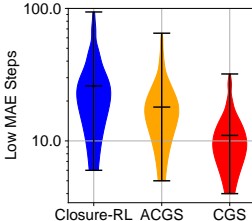

Figure 2: Results of Closure-RL applied to the advection equation. The relative MAEs are computed over 100 simulations with respect to (w.r.t.) the FGS. The shaded regions correspond to the respective standard deviations. The violin plot on the right shows the number of simulation steps until the relative MAE w.r.t. the FGS reaches the threshold of 1%.

# 4. Experiments

## 4.1. Advection Equation

First we apply Closure-RL to the 2D advection equation[4]:

$$\frac{\partial \psi}{\partial t} + u(x,y)\frac{\partial \psi}{\partial x} + v(x,y)\frac{\partial \psi}{\partial y} = 0, \tag{9}$$

where $\psi$ represents a physical concentration that is transported by the velocity field $C = (u,v)$. We employ periodic boundary conditions (PBCs) on the domain $\Omega = [0,1] \times [0,1]$.

For the FGS, this domain is discretized using $N_x = N_y = 256$ discretization points in each dimension. To guarantee stability, we employ a time step that ensures that the Courant–Friedrichs–Lewy (CFL) [43] condition is fulfilled. The spatial derivatives are calculated using central differences and the time stepping uses the fourth-order Runge-Kutta scheme [44]. The FGS is fourth order accurate in time and second order accurate in space.

We construct the CGS by employing the subsampling factors $d = 4$ and $d_t = 4$. Spatial derivatives in the CGS use an upwind scheme and time stepping is performed with the forward Euler method, resulting in first order accuracy in both space and time. All settings of numerical values used for the CGS and FGS are summarized in Table 2.

### 4.1.1. Initial Conditions

In order to prevent overfitting and promote generalization, we design the initializations of $\psi$, $u$ and $v$ to be different for each episode while still fulfilling the PBCs and guaranteeing the incompressibility of the velocity field. The velocity fields are sampled from a distribution $\mathcal{D}_{Train}^{Vortex}$ by taking a linear combination of Taylor-Greene vortices and an additional random translational field. Further details are provided in Appendix D.2.1. For visualization purposes, the concentration of a new episode is set to a random sample from the MNIST dataset [45] that is scaled to values in the range $[0,1]$. In order to increase the diversity of the initializations, we augment the data by performing random rotations of $\pm 90°$ in the image loading pipeline.

### 4.1.2. Training

We train the framework for a total of 2000 epochs and collect 1000 transitions in each epoch. More details regarding the training process as well as the training time are provided in Appendix D.
We note that the amount and length of episodes varies during the training process: The episodes are truncated based on the relative Mean Absolute Error (MAE) defined as $\text{MAE}(\psi^n, \tilde{\psi}^n) = \frac{1}{\bar{N}_x \cdot \bar{N}_y} \sum_{i,j} |\psi_{ij}^n - \tilde{\psi}_{ij}^n| / \psi_{max}$ between the CGS and FGS concentrations. Here, $\psi_{max}$ is the maximum observable value of the concentration $\psi$. If this error exceeds the threshold of $1.5\%$, the episode

---

[4]The code to reproduce all results in this paper can be found at `https://github.com/cselab/Closure-RL`.

is truncated. This ensures that during training, the CGS and FGS stay close to each other so that the reward signal is meaningful. As the agents become better during the training process, the mean episode length increases as the two simulations stay closer to each other for longer. We designed this adaptive training procedure in order to obtain stable simulations.

We note that inspired by [24] we experimented with adding an additional domain-specific global term to the reward (see Appendix E for details). We did not see any improvements compared to our Closure-RL formulation which indicates that the proposed reward structure is sufficient to learn an accurate closure model.

### 4.1.3. Accurate Coarse Grid Simulation

We also introduce the *'accurate coarse grid simulation'* (ACGS) in order to further compare the effecst of numerics and grid size in CGS and FGS. ACGS operates on the coarse grid, just like the CGS, with a higher order numerical scheme, than the CGS, so that it has the same order of accuracy as the FGS.

### 4.1.4. In- and Out-of-Distribution MAE

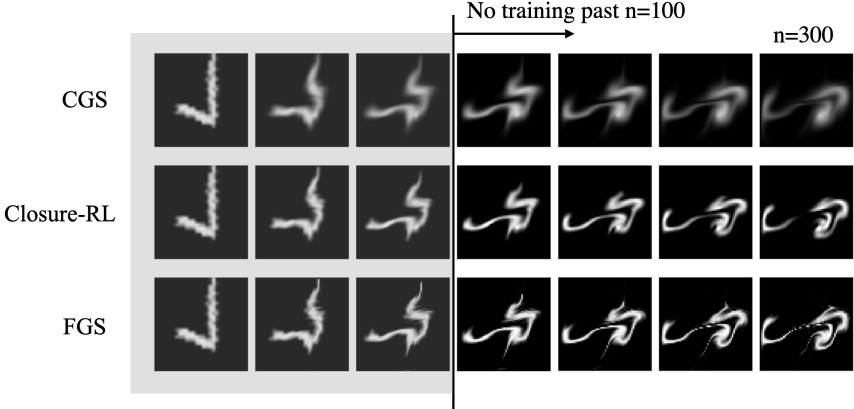

Figure 3: Example run comparing CGS, Closure-RL and FGS with the same ICs. The concentration is sampled from the test set and the velocity components are randomly sampled from $\mathcal{D}_{Train}^{Vortex}$. Note that the agents of the Closure-RLmethod have only been trained up to $n = 100$. However, they qualitatively improve the CGS simulation past that point.

We develop metrics for Closure-RL by running 100 simulations of 50 time steps each with different ICs. For the in-distribution case, the concentrations are sampled from the MNIST test set and the velocity fields are sampled from $\mathcal{D}_{Train}^{Vortex}$. To quantify the performance on out-of-distribution ICs, we also run evaluations on simulations using the Fashion-MNIST dataset (F-MNIST) [46] and a new distribution, $\mathcal{D}_{Test}^{Vortex}$, for the velocity fields. The latter is defined in Appendix D.2.2. The resulting error metrics of CGS, ACGS and Closure-RL w.r.t. the FGS are collected in Table 1. A qualitative example of the CGS, FGS and the Closure-RL method after training is presented in Figure 3. The example shows that Closure-RL is able to compensate for the dissipation that is introduced by the first order scheme and coarse grid in the CGS.

Closure-RL reduces the relative MAE of the CGS after 50 steps by $30\%$ or more in both in- and out-of-distribution cases. This shows that the agents have learned a meaningful correction for the truncation errors of the numerical schemes in the coarser grid. Closure-RL also outperforms the ACGS w.r.t. the MAE metric which indicates that the learned corrections emulate a higher-order scheme. This indicates that the proposed methodology is able to emulate the unresolved dynamics and is a suitable option for complementing existing numerical schemes.

**We note the strong performance of our framework w.r.t. the out-of-distribution examples.** For both unseen and out-of-distribution ICs as well as PDEPs, the framework was able to outperform

Table 1: Relative MAE at time step 50 averaged over 100 simulations with different ICs to both the velocity and concentration. All relative MAE values are averaged over the complete domain and reported in percent.

| Velocity Concentr. | $\mathcal{D}_{Train}^{Vortex}$ | | $\mathcal{D}_{Test}^{Vortex}$ | |
| --- | --- | --- | --- | --- |
| | MNIST | F-MNIST | MNIST | F-MNIST |
| CGS | $3.13 \pm 0.80$ | $3.23 \pm 0.92$ | $3.82 \pm 0.78$ | $3.12 \pm 0.73$ |
| ACGS | $1.90 \pm 0.51$ | $2.27 \pm 0.64$ | $2.28 \pm 0.47$ | $2.23 \pm 0.50$ |
| Closure-RL | $\mathbf{1.46 \pm 0.33}$ | $\mathbf{2.12 \pm 0.57}$ | $\mathbf{1.58 \pm 0.37}$ | $\mathbf{2.04 \pm 0.56}$ |
| Relative Improvements w.r.t. CGS | | | | |
| ACGS | -39% | -30% | -40% | -31% |
| Closure-RL | **-53%** | **-34%** | **-58%** | **-36%** |

CGS and ACGS. In our opinion, this indicates that we have discovered an actual model of the forcing terms that goes beyond the training scenarios.

### 4.1.5. Evolution of Numerical Error

The results in the previous sections are mostly focused on the difference between the methods after a rollout of 50 time steps. To analyze how the methods compare over the course of a longer rollout, we analyze the relative MAE at each successive step of a simulation with MNIST and $\mathcal{D}_{Train}^{Vortex}$ as distributions for the ICs. The results are shown in Figure 2.

The plots of the evolution of the relative error show that Closure-RL is able to improve the CGS for the entire range of a 400-step rollout, although it has only been trained for 100 steps. This implies that the agents are seeing distributions of the concentration that have never been encountered during training and are able to generalize to these scenarios. When measuring the duration of simulations for which the relative error stays below $1\%$, we observe that the Closure-RL method outperforms both ACGS and CGS, indicating that the method is able to produce simulations with higher long term stability than CGS and ACGS. We attribute this to our adaptive scheme for episode truncation during training as introduced in Section 4.1.2 and note that the increased stability can be observed well beyond the training regime.

### 4.1.6. Interpretation of Actions

The intention of our closure scheme is to negate the errors introduced by the numerical scheme. For the advection CGS, we are able to show that the actions taken are indeed highly correlated with the optimal action based on the errors of the numerical scheme used. In Appendix G, we are reporting the derivation of this optimal action as well as the correlation with the actions actually taken by Closure-RL. The obtained correlation coefficients are between $0.7 - 0.82$ depending on the task.

### 4.2. Burgers' Equation

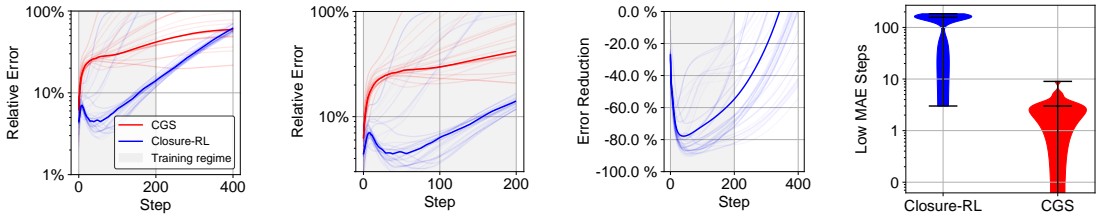

Figure 4: Results of Closure-RL applied to the Burgers' equation. The relative MAEs are computed over 100 simulations with respect to the FGS. The shaded regions correspond to the respective standard deviations. The violin plot on the right shows the number of simulation steps until the relative MAE w.r.t. the FGS reaches the threshold of $10\%$.

As a second example, we apply our framework to the 2D viscous Burgers' equation:

$$\frac{\partial \boldsymbol{\psi}}{\partial t} + (\boldsymbol{\psi} \cdot \nabla)\boldsymbol{\psi} - \nu \nabla^2 \boldsymbol{\psi} = 0. \tag{10}$$

Here, $\boldsymbol{\psi} := (u, v)$ consists of both velocity components and the PDE has the single input PDEP $C = \nu$. As for the advection equation, we assume periodic boundary conditions on the domain $\Omega = [0, 1] \times [0, 1]$. In comparison to the advection example, we are now dealing with two solution variables and thus $k = 2$.

For the FGS, the aforementioned domain is discretized using $N_x = N_y = 150$ discretization points in each dimension. Moreover, we again choose $\Delta t$ to fulfill the CFL condition for stability (see Table 3). The spatial derivatives are calculated using the upwind scheme and the forward Euler method is used for the time stepping [44]. We construct the CGS by employing the subsampling factors $d = 5$ and $d_t = 10$. For the Burgers experiment, we apply a mean filter $K$ with kernel size $d \times d$ before the actual subsampling operation. The mean filter is used to eliminate higher frequencies in the fine grid state variables, which would lead to accumulating high errors. The CGS employs the same numerical schemes as the FGS here. This leads to first order accuracy in both space and time. All numerical settings used for the CGS and FGS are collected in Table 3. In this example, where FGS and CGS are using the same numerical scheme, the Closure-RL framework has to focus solely on negating the effects of the coarser discretization. For training and evaluation, we generate random, incompressible velocity fields as ICs (see Appendix D.2.1 for details) and set the viscosity $\nu$ to $0.003$. We observed that the training of the predictions actions can be improved by multiplying the predicted forcing terms with $\widetilde{\Delta t}$. This is consistent with our previous analysis in Appendix G as the optimal action is also multiplied but this factor. Again, we train the model for 2000 epochs with 1000 transitions each. The maximum episode length during training is set to 200 steps and, again, we truncate the episodes adaptively, when the relative error exceeds $20\%$. Further details on training Closure-RL for the Burgers' equation can be found in Appendix D. Visualization of the results can be found in Appendix C.2 and the resulting relative errors w.r.t. the FGS are shown in Figure 4. The Closure-RL method again improves the CGS significantly, also past the point of the 200 steps seen during training. Specifically, in the range of step $0$ to step $100$ in which the velocity field changes fastest, we see a significant error reductions up to $-80\%$. When analyzing the duration for which the episodes stay under the relative error of $10\%$, we observe that the mean number of steps is improved by two order of magnitude, indicating that the method is able to improve the long term accuracy of the CGS.

## 5. Conclusions

We propose Closure-RL, a novel framework for the automated discovery of closure models of coarse grained discretizations of time-dependent PDEs. It utilizes a grid-based RL formulation with a FCN for both the policy and the value network. This enables the incorporation of local rewards without necessitating individual neural networks for each agent and allows to efficiently train a large number of agents. Moreover, the framework trains on rollouts without needing to backpropagate through the numerical solver itself. We show that Closure-RL develops a policy that compensates for numerical errors in a CGS of both the 2D advection and Burgers' equation. More importantly we find that the learned closure model can be used for predictions in extrapolative test cases.

Further work may focus on extending the formulation to irregular grids by using a Graph Convolutional Network [47] instead of the FCN. Similarly to the current approach, we can place one agent at each discretization point whose receptive field now depends on the connection between the graph nodes. Moreover, for very large systems of interest the numerical schemes used are often multi-grid and the grid-based RL framework should reflect this. For such cases, we suggest defining separate rewards for each of the grids employed by the numerical solver. Additionally, Closure-RL may serve as a stepping-stone towards incorporating further inductive bias or constraints as its action space can readily be adapted. For instance, the actions for some agents could be explicitly parameterized or coupled with their neighboring agents. In this regard, Closure-RL could also be used to further improve other, domain-specific closure models.

# Acknowledgements

S.K. and P.K. acknowledge support by The European High Performance Computing Joint Undertaking (EuroHPC) Grant DCoMEX (956201-H2020-JTI-EuroHPC-2019-1) and support by the Defense Advanced Research Projects Agency (DARPA) through Award HR00112490489.

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

## A. Neural Network Architecture

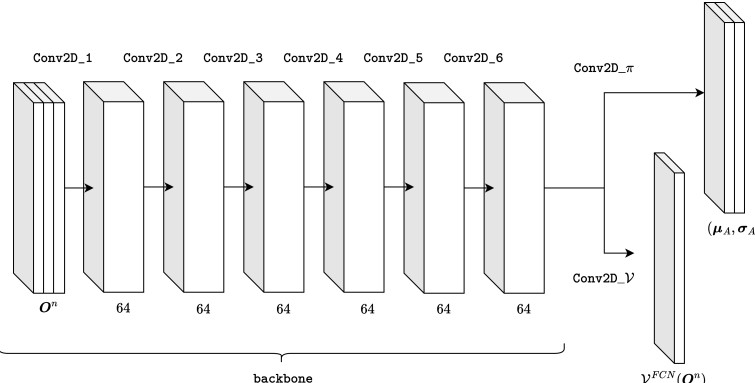

Figure 5: The IRCNN backbone takes the current global observation and maps it to a feature tensor. This feature tensor is passed into two different convolutional layers that predict the per-discretization point action-distribution-parameters and state-values. In the case of a PDE with three spatial dimensions, the architecture would need to be based on three-dimensional convolutional layers instead.

The optimal policy is expected to compensate for errors introduced by the numerical method and its implementation on a coarse grid. We use the Image Restoration CNN (IRCNN) architecture proposed in [48] as our backbone for the policy- and value-network. A discussion of this choice can be found in Appendix F. In Figure 5 we present an illustration of the architecture and show that the policy- and value network share the same backbone. The policy- and value network only differ by their last convolutional layer, which takes the features extracted by the backbone and maps them either to the action-distribution-parameters $\boldsymbol{\mu}_A \in \mathcal{A}$ and $\boldsymbol{\sigma}_A \in \mathcal{A}$ or the predicted return value for each agent. The local policies are assumed to be independent, such that we can write the distribution at a specific discretization point as

$$\pi_{ij}(\boldsymbol{A}_{ij}|\boldsymbol{O}_{ij}) = \mathcal{N}(\boldsymbol{\mu}_{A,ij}, \boldsymbol{\sigma}_{A,ij}). \tag{11}$$

During training, the actions are sampled to allow for exploration, and during inference only the mean is taken as the action of the agent.
We note that the padding method used for the FCN can incorporate boundary conditions into the architecture. For instance, in the case of periodic boundary conditions, we propose to use circular padding that involves wrapping around values from one end of the input tensor to the other.

---

**Algorithm 1** Adapted PPO Algorithm

---

**Input:** Initial policy weights $\theta_1$, initial value function weights $\phi_1$,
clip ratio $\epsilon$, discount factor $\gamma$, entropy regularization weight $\beta$
**for** $k = 1, 2, \ldots$ **do**
    Collect set of trajectories $D_k$ with the global policy $\Pi_{\theta_k}^{FCN}$
    *# Update global policy*
    Update $\theta$ by performing a SGD step on
    $\theta_{k+1} = \arg\max_\theta \frac{1}{\|D_k\|} \sum_{\boldsymbol{O}^n, \boldsymbol{A}^n \in D_k} L_\Pi(\boldsymbol{O}^n, \boldsymbol{A}^n, \theta_k, \theta, \beta)$
    *# Update global value network*
    Compute returns for each transition using $\boldsymbol{G}^n = \sum_{i=n}^{N} \gamma^{i-t} \boldsymbol{R}^i$ where $N$ is the length of the respective trajectory
    Update $\phi$ by performing a SGD step on $\phi_{k+1} = \arg\min_\phi \frac{1}{\|D_k\|} \sum_{\boldsymbol{O}^n, \boldsymbol{G}^n \in D_k} L_\mathcal{V}(\boldsymbol{O}^n, \boldsymbol{G}^n, \phi)$
**end for**

---

## B. Adapted PPO Algorithm

As defined in Section 3.5 the loss function for the global value function is

$$L_\mathcal{V}(\boldsymbol{O}^n, \boldsymbol{G}^n, \phi) = ||\mathcal{V}_\phi^{FCN}(\boldsymbol{O}^n) - \boldsymbol{G}^n||_2^2.$$

We note that this notation contains the weights $\phi$, which parameterize the underlying neural network.

The objective for the global policy is defined as

$$L_\Pi(\boldsymbol{O}^n, \boldsymbol{A}^n, \theta_k, \theta, \beta) := \frac{1}{\tilde{N}_x \cdot \tilde{N}_y} \sum_{i,j=1}^{\tilde{N}_x, \tilde{N}_y} L_{\pi_{ij}}(\boldsymbol{O}_{ij}^n, \boldsymbol{A}_{ij}^n, \theta_k, \theta) - \beta H(\pi_{ij,\theta}),$$

where $H(\pi_{ij,\theta})$ is the entropy of the local policy and $L_{\pi_{ij}}$ is the standard single-agent PPO-Clip objective

$$L_{\pi_{ij}}(o, a, \theta_k, \theta) = \min\left( \frac{\pi_{ij,\theta}(a|o)}{\pi_{ij,\theta_k}(a|o)} Adv^{\pi_{ij,\theta_k}}(o, a), \text{clip}\left( \frac{\pi_{ij,\theta}(a|o)}{\pi_{ij,\theta_k}(a|o)}, 1 - \epsilon, 1 + \epsilon \right) Adv^{\pi_{ij,\theta_k}}(o, a) \right).$$

The advantage estimates $Adv^{\pi_{ij,\theta_k}}$ are computed with generalized advantage estimation (GAE) [49] using the output of the global value network $\mathcal{V}_\phi^{FCN}$.

The resulting adapted PPO algorithm is presented in Algorithm 1. Major differences compared to the original PPO algorithm are vectorized versions of the value network loss and PPO-Clip objective, as well as a summation over all the discretization points of the domain before performing an update step.

# C. Additional Results

## C.1. Advection Equation

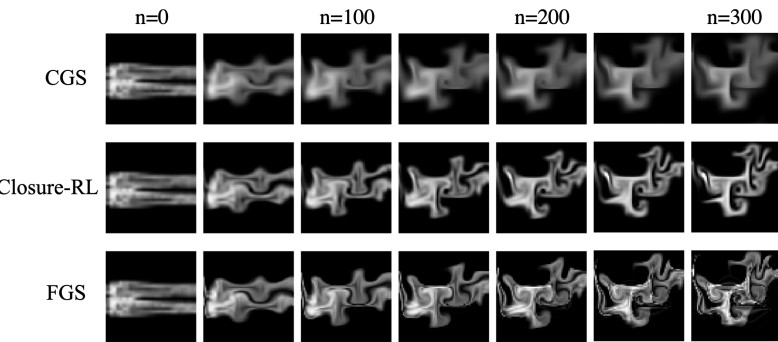

Figure 6: $\psi^0$ is sampled from the MNIST test set. The velocity field is sampled from $\mathcal{D}_{Test}^{Vortex}$ (See Appendix D.2.1). Here, the IC of the concentration comes from a different distribution than the one used for training.

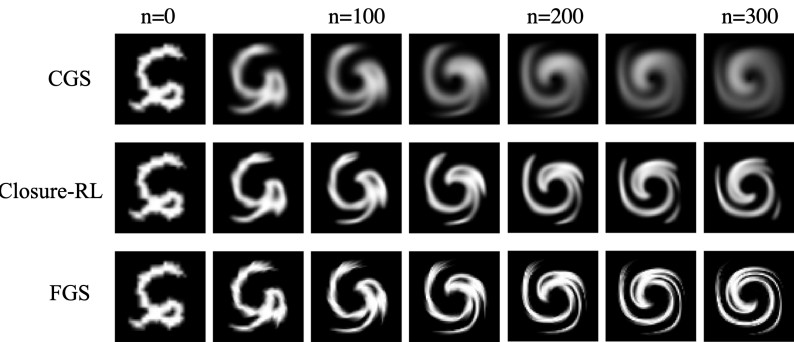

Figure 7: $\psi^0$ is a sample from the F-MNIST test set. The velocity field is sampled from $\mathcal{D}_{Train}^{Vortex}$ (See Appendix D.2.2). Note that the velocity field comes from a different distribution than the one used for training.

## C.2. Burgers' Equation

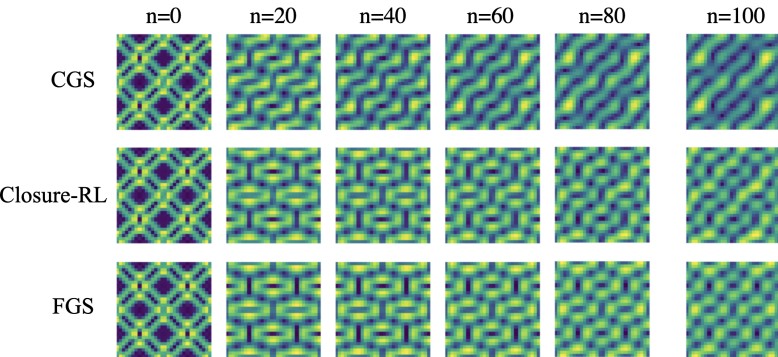

Figure 8: $\psi^0$ is a sample from $\mathcal{D}_{Train}^{Vortex}$. We plot the velocity magnitude of every 20th step of simulation with 100 coarse time steps. The example shows that Closure-RL does also qualitatively keep the simulation closer to the FGS. The failure of the CGS to account for the subgrid-scale dynamics leads to diverging trajectories.

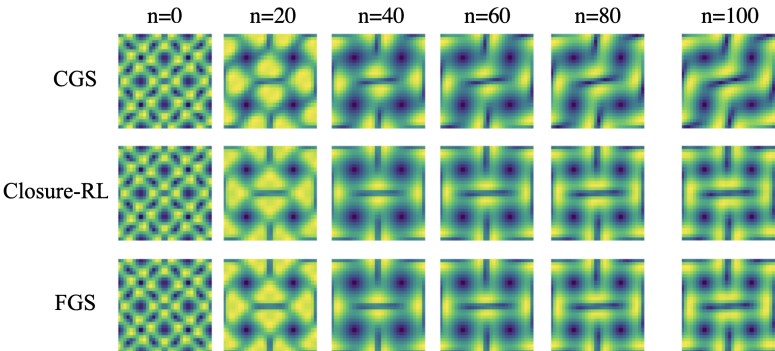

Figure 9: Velocity magnitude plotted for a 100-step roll out. The set-up is the same as in Figure 8 and the example again shows, that Closure-RL leads to qualitatively different dynamics during a roll-out that are closer to those of the FGS.

# D. Technical Details on Hyperparamters and Training Runs

During training, we use entropy regularization in the PPO objective with a factor of $0.1$ and $0.05$ for the advection and Burgers' equation respectively to encourage exploration. The discount factor is set to $0.95$ and the learning rate to $1 \cdot 10^{-5}$. Training is done over 2000 epochs. In each epoch, 1000 transitions are collected. One policy network update is performed after having collected one new episode. We use a batch size of 10 for training. The total number of trainable weights amounts to $188, 163$ and the entire training procedure took about 8 hours for the advection equation on an Nvidia A100 GPU. For the Burgers' equation, training took about $30$ hours on the same hardware. We save the policy every 50 epochs and log the corresponding MAE between CGS and FGS after 50 time steps. For evaluation on the advection equation, we chose the policy from epoch 1500 because it had the lowest logged MAE value. Figure 10 and Figure 11 show the reward curves and evolutions of episode lengths. As expected, the episode length increases as the agents become better at keeping the CGS and FGS close to each other.

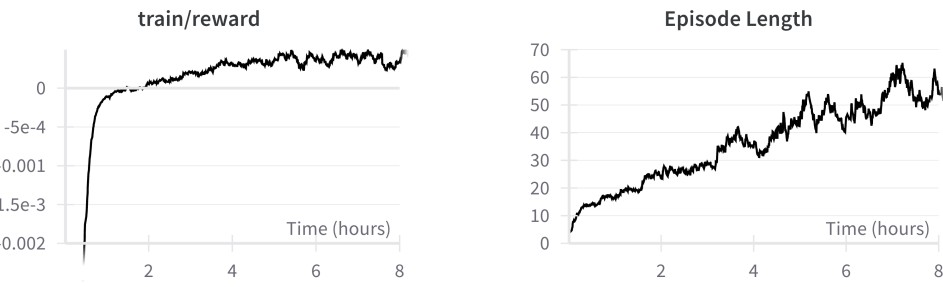

Figure 10: Visualizations of the evolution of the reward metric averaged over the agents and episode length during training on the advection equation.

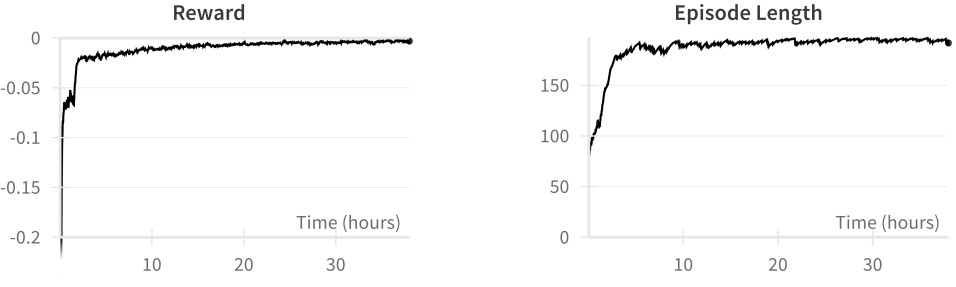

Figure 11: Visualizations of the evolution of the reward metric averaged over the agents and episode length during training on the Burgers' equation.

Table 2: Numerical values used for the advection CGS and FGS. Note that $\widetilde{\Delta t}$ is chosen to guarantee that the CFL condition in the CGS is fulfilled.

| | | |
|---|---|---|
| $\Omega$ | $[0,1] \times [0,1]$ | |
| $\tilde{N}_x, \tilde{N}_y$ | 64 | |
| $d, d_t$ | 4 | |
| **Resulting other values:** | | |
| $N_x, N_y$ | $d \cdot \tilde{N}_x, d \cdot \tilde{N}_y$ | $= 256$ |
| $\Delta x, \Delta y$ | $1/N_x, 1/N_y$ | $\approx 0,0039$ |
| $\widetilde{\Delta x}, \widetilde{\Delta y}$ | $1/\tilde{N}_x, 1/\tilde{N}_y$ | $\approx 0,0156$ |
| $\widetilde{\Delta t}$ | $0.9 \cdot \min(\widetilde{\Delta x}, \widetilde{\Delta y})$ | $\approx 0,0141$ |
| $\Delta t$ | $\widetilde{\Delta t}/d_t$ | $\approx 0,0035$ |
| **Discretization schemes:** | | |
| FGS, Space | Central difference | |
| FGS, Time | Fourth-order Runge-Kutta | |
| CGS, Space | Upwind | |
| CGS, Time | Forward Euler | |

Table 3: Numerical values used for the Burgers' CGS and FGS. Again, $\widetilde{\Delta t}$ is chosen to guarantee that the CFL condition in the CGS is fulfilled.

| | | |
|---|---|---|
| $\Omega$ | $[0,1] \times [0,1]$ | |
| $\tilde{N}_x, \tilde{N}_y$ | 30 | |
| $d, d_t$ | 5, 10 | |
| **Resulting other values:** | | |
| $N_x, N_y$ | $d \cdot \tilde{N}_x, d \cdot \tilde{N}_y$ | $= 150$ |
| $\Delta x, \Delta y$ | $1/N_x, 1/N_y$ | $\approx 0,0067$ |
| $\widetilde{\Delta x}, \widetilde{\Delta y}$ | $1/\tilde{N}_x, 1/\tilde{N}_y$ | $\approx 0,0333$ |
| $\widetilde{\Delta t}$ | $0.9 \cdot \min(\widetilde{\Delta x}, \widetilde{\Delta y})$ | $= 0,03$ |
| $\Delta t$ | $\widetilde{\Delta t}/d_t$ | $= 0,003$ |
| **Discretization schemes:** | | |
| FGS, Space | Upwind | |
| FGS, Time | Forward Euler | |
| CGS, Space | Upwind | |
| CGS, Time | Forward Euler | |

## D.1. Receptive Field of FCN

In our Closure-RL problem setting, the receptive field of the FCN corresponds to the observation $O_{ij}$ the agent at point $(i, j)$ is observing. In order to gain insight into this, we analyze the receptive field of our chosen architecture.

In the case of the given IRCNN architecture, the size of the receptive field (RF) of layer $i$ can be recursively calculated given the RF of layer afterward with

$$\text{RF}_{i+1} = \text{RF}_i + (\text{Kernel Size}_{i+1} - 1) \cdot \text{Dilation}_{i+1} \tag{12}$$
$$= \text{RF}_i + 2 \cdot \text{Dilation}_{i+1}. \tag{13}$$
$$\tag{14}$$

The RF field of the first layer $\text{RF}_1$ is equal to its kernel size. By then using the recursive rule, we can calculate the RF at each layer and arrive at a value of $\text{RF}_7 = 33$ for the entire network. From this, we

now arrive at the result that agent $(i, j)$ sees a $33 \times 33$ patch of the domain centered around its own location.

Table 4: Hyperparameters of each of the convolutional layers of the neural network architecture used for the advection equation experiment and the resulting receptive field (RF) at each layer. For the Burgers' equation experiment the architecture is simply adapted by setting the number of in channels of `Conv2D_1` to 2 and the number of out channels of `Conv2D_π` to 4.

| Layer | In Channels | Out Channels | Kernel | Padding | Dilation | RF |
|---|---|---|---|---|---|---|
| `Conv2D_1` | 3 | 64 | 3 | 1 | 1 | 3 |
| `Conv2D_2` | 64 | 64 | 3 | 2 | 2 | 7 |
| `Conv2D_3` | 64 | 64 | 3 | 3 | 3 | 13 |
| `Conv2D_4` | 64 | 64 | 3 | 4 | 4 | 21 |
| `Conv2D_5` | 64 | 64 | 3 | 3 | 3 | 27 |
| `Conv2D_6` | 64 | 64 | 3 | 2 | 2 | 31 |
| `Conv2D_π` | 64 | 2 | 3 | 1 | 1 | 33 |
| `Conv2D_V` | 64 | 1 | 3 | 1 | 1 | 33 |

## D.2. Diverse Velocity Field Generation

### D.2.1. Distribution for Training

For the advection equation experiment, the velocity field is randomly generated by taking a linear combination of Taylor-Greene vortices and an additional random translational field. Let $\boldsymbol{u}_{ij}^{TG,k}, \boldsymbol{v}_{ij}^{TG,k}$ be the velocity components of the Taylor Greene Vortex with wave number $k$ that are defined as

$$\boldsymbol{u}_{ij}^{TG,k} := \cos(k\boldsymbol{x}_i) \cdot \sin(k\boldsymbol{y}_j) \tag{15}$$

$$\boldsymbol{v}_{ij}^{TG,k} := -\sin(k\boldsymbol{x}_i) \cdot \cos(k\boldsymbol{y}_h). \tag{16}$$

Furthermore, define the velocity components of a translational velocity field as $u^{TL}, v^{TL} \in \mathbb{R}$. To generate a random incompressible velocity field, we sample 1 to 4 $k$'s from the set $\{1, ..., 6\}$. For each $k$, we also sample a $\text{sign}_k$ uniformly from the set $\{-1, 1\}$ in order to randomize the vortex directions. For an additional translation term, we sample $u^{TL}, v^{TL}$ independently from uniform$(-1, 1)$. We then initialize the velocity field to

$$\boldsymbol{u}_{ij} := u^{TL} + \sum_k \text{sign}_k \cdot \boldsymbol{u}_{ij}^{TG,k} \tag{17}$$

$$\boldsymbol{v}_{ij} := v^{TL} + \sum_k \text{sign}_k \cdot \boldsymbol{v}_{ij}^{TG,k}. \tag{18}$$

We will refer to this distribution of vortices as $\mathcal{D}_{Train}^{Vortex}$.

For the Burgers' equation experiment, we make some minor modifications to the sampling procedure. The sampling of translational velocity components is omitted and 2 to 4 $k$'s are sampled from $\{2, 4, 6, 8\}$. The latter ensures that the periodic boundary conditions are fulfilled during initialization which is important for the stability of the simulations.

### D.2.2. Distribution for Testing

First, a random sign sign is sampled from the set $\{-1, 1\}$. Subsequently, a scalar $a$ is randomly sampled from a uniform distribution bounded between 0.5 and 1. The randomization modulates both the magnitude of the velocity components and the direction of the vortex, effectively making the field random yet structured. The functional forms of $\boldsymbol{u}_{ij}^C$ and $\boldsymbol{v}_{ij}^C$ are then expressed as

$$\boldsymbol{u}_{ij}^V := \text{sign} \cdot a \cdot \sin^2(\pi\boldsymbol{x}_i)\sin(2\pi\boldsymbol{y}_j) \tag{19}$$

$$\boldsymbol{v}_{ij}^V := -\text{sign} \cdot a \cdot \sin^2(\pi\boldsymbol{y}_j)\sin(2\pi\boldsymbol{x}_i). \tag{20}$$

In the further discussion, we will refer to this distribution of vortices as $\mathcal{D}_{Test}^{Vortex}$.

Table 5: Runtime of one simulation step in ms of the different simulations averaged over 500 steps.

|  | CGS | ACGS | Closure-RL | FGS |
|---|---|---|---|---|
| Advection | $0.31 \pm 0.00$ | $0.96 \pm 0.00$ | $2.66 \pm 0.96$ | $89.52 \pm 0.47$ |
| Burgers' | $0.25 \pm 0.00$ | - | $1.82 \pm 0.01$ | $10.16 \pm 0.03$ |

## D.3. Computational Complexity

To quantitatively compare the execution times of the different simulations, we measure the runtime of performing one update step of the environment and report them in Table 5. As expected, Closure-RL increases the runtime of the CGS. However, it stays below the FGS times by at least a factor of $5$. The difference is especially pronounced in the example of the advection equation, where the FGS uses a high order scheme on a fine grid, which leads to an execution time difference between Closure-RL and FGS of more than an order of magnitude.

We additionally profiled our code for the advection experiment and found that 71.3% of the runtime is spent on obtaining trajectories from the FGS simulation. Only 1.3 % of the time is spent on the CGS (excluding the forward pass through the FCN), highlighting the significant computational overhead of the finer grid. The forward passes through the neural network account for 5.0% of the time, while updating the policy takes another 5.5%. This is in agreement with the theoretical considerations above that show that the FGS is computationally more expensive than the CGS. We note that there is room for optimization of the numerical solution, as currently the generation of the FGS simulation is not parallelized.

# E. Addition of a Global Reward

To investigate the effect of adding a global problem specific reward to the reward function, we repeated our experiment for the advection equation and added a penalty based on the deviation from the conservation of the total mass to each local reward term. The deviation from the total mass was scaled so that both parts of the reward term had the same order of magnitude. The obtained mean absolute error reduction compared to CGS was approximately 52 percent after 50 time steps during predictions (the local-only reward resulted in 53 percent error reduction) and thus did not offer any additional benefits compared to our original formulation. The situation may require further investigation for different global constraints and different PDEs. However for the present study the local rewards seem to be sufficient in order to obtain good closures.

# F. Discussion of Architecture Choice

We have chosen the IRCNN architecture [48] to have a small receptive field such that our agents learn and act locally in a manner that is reminiscent of the numerical discretizations of PDEs. The IRCNN consists of multiple convolutional layers and has been shown to be very successful in image restoration tasks [9]. Architectures such as LoFi [50], which also only have a small receptive field, could be a possible architecture choice for Closure-RL as well, although this would require further investigations.

Other architectures such as U-Nets [51] or Vision-Transformers [52–54] are desired to capture multiscale features or long-range dependencies which would significantly increase the receptive field of each agent. Based on our studies regarding the interpretation of actions (see Appendix G) as well as the derivation of the theoretical optimal action for one of the cases explored in detail (see Section Appendix H), these dependencies are generally not needed, as local information should be sufficient to complement numerical schemes. We note that this is not necessarily the case for other systems of interest such as PDEs with forcing terms, PDEs with time-varying parameters or Integro-Differential Equations.

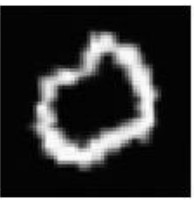 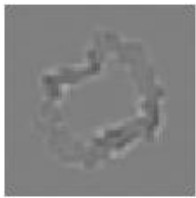 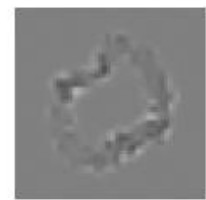 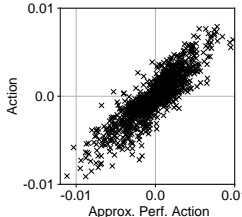

Figure 12: Visual comparison of the input concentration $\tilde{\psi}^n$, taken actions $\boldsymbol{A}^n$ and the corresponding numerical approximation of the optimal action from left to right. The figure on the right-hand side qualitatively shows the linear relation between approximated optimal actions and taken actions.

# G. Interpretation of Actions

We are able to derive the optimal update rule for the advection CGS that negates the errors introduced by the numerical scheme (see Appendix H for the derivation):

$$\tilde{\psi}^{n+1} = \mathcal{G}(\tilde{\psi}^n - \widetilde{\Delta t}\tilde{\epsilon}^{n-1}, \tilde{C}^n), \tag{21}$$

Here, $\tilde{\epsilon}^{n-1} \in \mathbb{R}^{\tilde{N}_x \times \tilde{N}_y}$ is the truncation error of the previous step. However, note that there exists no closed-form solution to calculate the truncation error $\tilde{\epsilon}^{n-1}$ if only the state of the simulation is given. We therefore employ a numerically approximate for the truncation errors for further analysis:

$$\tilde{\epsilon}^n = \frac{\widetilde{\Delta t}}{2}\tilde{\psi}^n_{tt} + |\tilde{u}^n|\frac{\widetilde{\Delta x}}{2}\tilde{\psi}^n_{xx} + |\tilde{v}^n|\frac{\widetilde{\Delta y}}{2}\tilde{\psi}^n_{yy} + \mathcal{O}(\widetilde{\Delta t}^2, \widetilde{\Delta x}^2, \widetilde{\Delta y}^2) \tag{22}$$

Here $\tilde{\psi}^n_{xx}, \tilde{\psi}^n_{yy}$ and $\tilde{\psi}^n_{tt}$ are second derivatives of $\psi$ that are numerically estimated using second-order central differences.

We compare the obtained optimal update rule for the CGS with the predicted mean action $\boldsymbol{\mu}_A$ of the policy and thus the learned actions. Figure 12 visualizes an example, which indicates that there is a strong linear relationship between the predicted mean action and the respective numerical estimate of the optimal action.

In order to further quantify the similarity between the numerical estimates of $-\widetilde{\Delta t}\tilde{\epsilon}^{n-1}_{i,j}$ and the taken action, we compute the Pearson product-moment correlation coefficient for 100 samples. The results are presented in Table 6 and show that the learned actions as well as the optimal action of the CGS are highly correlated for all different combinations of seen and unseen ICs and PDEPs.

Table 6: Mean and standard deviation of Pearson product-moment correlation coefficients between $\boldsymbol{\mu}_A^n$ and numerically approximated $-\widetilde{\Delta t}\tilde{\epsilon}^{n-1}$ for 100 samples for different combinations of velocity fields and initializations of the concentration.

|  | MNIST | F-MNIST |
|---|---|---|
| $\mathcal{D}^{Vortex}_{Train}$ | $0.82 \pm 0.05$ | $0.70 \pm 0.08$ |
| $\mathcal{D}^{Vortex}_{Test}$ | $0.82 \pm 0.03$ | $0.72 \pm 0.08$ |

Additionally, we note that the truncation error contains a second-order temporal derivative. At first glance, it might seem surprising that the model would be able to predict this temporal derivative as the agents can only observe the current time step. However, the observation contains both the PDE solution, i.e. the concentration for the present examples, as well as the PDEPs, i.e. the velocity fields. Thus, both the concentration and velocity field are passed into the FCN and due to the velocity fields enough information is present to infer this temporal derivative.

# H. Proof: Theoretically Optimal Action

To explore how the actions of the agents can be interpreted, we analyze the optimal actions based on the used numerical schemes. The *perfect* solution would fulfill

$$\mathcal{M}(\tilde{\boldsymbol{\psi}}^n) + \tilde{\boldsymbol{\epsilon}}^n = 0.$$

Here, $\mathcal{M}$ is the numerical approximation of the PDE an $\tilde{\epsilon}^n$ is the truncation error.

We refer to the numerical approximations of the derivatives in the CGS as $\mathcal{T}^{FE}$ for forward Euler and $\mathcal{D}^{UW}$ for the upwind scheme. We obtain

$$0 = \mathcal{M}(\tilde{\boldsymbol{\psi}}^n) + \tilde{\boldsymbol{\epsilon}}^n \tag{23}$$

$$= \mathcal{T}^{FE}(\tilde{\boldsymbol{\psi}}^n, \tilde{\boldsymbol{\psi}}^{n+1}) + \tilde{\boldsymbol{u}}^n \mathcal{D}_x^{UW}(\tilde{\boldsymbol{\psi}}^n) + \tilde{\boldsymbol{v}}^n \mathcal{D}_y^{UW}(\tilde{\boldsymbol{\psi}}^n) + \tilde{\boldsymbol{\epsilon}}^n \tag{24}$$

$$= \frac{\tilde{\boldsymbol{\psi}}^{n+1} - \tilde{\boldsymbol{\psi}}^n}{\widetilde{\Delta t}} + \tilde{\boldsymbol{u}}^n \mathcal{D}_x^{UW}(\tilde{\boldsymbol{\psi}}^n) + \tilde{\boldsymbol{v}}^n \mathcal{D}_y^{UW}(\tilde{\boldsymbol{\psi}}^n) + \tilde{\boldsymbol{\epsilon}}^n. \tag{25}$$

$$\tag{26}$$

By rewriting, we obtain the following time stepping rule

$$\tilde{\boldsymbol{\psi}}^{n+1} = \tilde{\boldsymbol{\psi}}^n - \widetilde{\Delta t}\left(\tilde{\boldsymbol{u}}^n \mathcal{D}_x^{UW}(\tilde{\boldsymbol{\psi}}^n) + \tilde{\boldsymbol{v}}^n \mathcal{D}_y^{UW}(\tilde{\boldsymbol{\psi}}^n) + \tilde{\boldsymbol{\epsilon}}^n\right) \tag{27}$$

$$= \tilde{\boldsymbol{\psi}}^n - \widetilde{\Delta t}\left(\tilde{\boldsymbol{u}}^n \mathcal{D}_x^{UW}(\tilde{\boldsymbol{\psi}}^n) + \tilde{\boldsymbol{v}}^n \mathcal{D}_y^{UW}(\tilde{\boldsymbol{\psi}}^n)\right) - \widetilde{\Delta t}\tilde{\boldsymbol{\epsilon}}^n \tag{28}$$

$$= \mathcal{G}(\tilde{\boldsymbol{\psi}}^n, \tilde{C}^n) - \widetilde{\Delta t}\tilde{\boldsymbol{\epsilon}}^n. \tag{29}$$

where $\tilde{C}^n := (\tilde{\boldsymbol{u}}^n, \tilde{\boldsymbol{v}}^n)$. This update rule would theoretically find the *exact* solution. It involves a coarse step followed by an additive correction after each step. We can define $\tilde{\boldsymbol{\phi}}^{n+1} := \tilde{\boldsymbol{\psi}}^{n+1} + \widetilde{\Delta t}\tilde{\boldsymbol{\epsilon}}^n$, to bring the equation into the same form as seen in the definition of the RL environment

$$\tilde{\boldsymbol{\phi}}^{n+1} = \mathcal{G}(\tilde{\boldsymbol{\phi}}^n - \widetilde{\Delta t}\tilde{\boldsymbol{\epsilon}}^{n-1}, \tilde{C}^n),$$

which illustrates that the optimal action at step $n$ would be the previous truncation error times the time increment

$$\boxed{\boldsymbol{A}^{n*} = \widetilde{\Delta t}\tilde{\boldsymbol{\epsilon}}^{n-1}.} \tag{30}$$

