# OpenReview forum: "Closure Discovery for Coarse-Grained Partial Differential Equations Using Grid-based Reinforcement Learning"
_CPAL.cc/2025/Proceedings_Track — CPAL 2025 (Proceedings Track) Oral_

### Official Review · Reviewer_zz5V · 2025-01-09
**Closure Discovery of coarse grained discretizations of time-dependent PDEs using Reinforcement Learning**

**Rating:** 7
**Confidence:** 3

**Review:**

This paper presents Closure-RL, a framework for discovering closures for coarse-grained discretizations of time-dependent PDEs. The approach employs grid-based reinforcement learning with a CNN serving as the policy and value network. Notably, this framework enables the incorporation of local rewards without the need for separate neural networks for each agent, facilitating the efficient training of many agents. The authors demonstrate the effectiveness of the proposed method through experiments on 2D advection and Burgers’ equations.

The proposed framework seems novel and the experiments are compelling, with additional supporting details provided in the Appendix. However, addressing the following points would further enhance the quality of the manuscript:

1. The authors use an IRCNN architecture for the policy network. While IRCNN is a single-scale CNN with a localized and relatively small receptive field, it would be insightful to explore the impact of multi-scale CNNs such as U-Net [1], which has larger receptive fields. Alternatively, vision transformers like SwinIR [2], capable of capturing long-range dependencies, could be investigated. For a more detailed discussion on the influence of receptive fields on model performance, please refer to [3].

2. Certain sections of the manuscript can be improved in clarity and organization:
- Line 54 seems incomplete!
- Some paragraphs are overly lengthy, making them harder to follow. For example, the first two paragraphs of the Introduction and the paragraph before the Conclusion could be split for better readability.
- Section 3 is hard to follow. Including an example of a specific PDE could greatly enhance the reader's understanding.
- Equation 2 needs more explanation.

[1] Ronneberger, Olaf, Philipp Fischer, and Thomas Brox. "U-net: Convolutional networks for biomedical image segmentation." Medical image computing and computer-assisted intervention–MICCAI 2015: 18th international conference, Munich, Germany, October 5-9, 2015, proceedings, part III 18. Springer International Publishing, 2015.
[2] Liang, Jingyun, et al. "Swinir: Image restoration using swin transformer." Proceedings of the IEEE/CVF international conference on computer vision. 2021.
[3] Khorashadizadeh, AmirEhsan, et al. "LoFi: Neural Local Fields for Scalable Image Reconstruction" arXiv preprint arXiv:2411.04995 (2024).

---

### Official Review · Reviewer_GU4Y · 2025-01-11
**Closure-RL introduces a novel RL-based approach for improving coarse-grained PDE simulations with good accuracy and efficiency gains**

**Rating:** 7
**Confidence:** 3

**Review:**

CPAL 76

The paper introduces Closure-RL, a grid-based reinforcement learning framework designed to discover closures for coarse-grained PDEs. By using a fully convolutional network, the method improves coarse-grid simulation accuracy while maintaining computational efficiency. Closure-RL shows strong performance on the 2D advection and Burgers’ equations, achieving significant reductions in numerical errors and better generalization to out-of-distribution scenarios.

I rate this paper quite highly, because personally, I believe the paper is well structured, with clear motivation and strong performance. There are no obvious weaknesses in this paper to me. One of my main interests is to see how will the Closure-RL extend to irregular unstructured grids.

---

### Official Review · Reviewer_RVxK · 2025-01-11
**Interesting paper with solid experiments**

**Rating:** 8
**Confidence:** 3

**Review:**

I think overall it's an interesting paper. I particularly like that it takes inspiration from RL for image reconstruction and based on that develops the framework for PDEs. The experiments also look solid: The method is tested in multiple environments, and both the qualitative results and quantitative results of Closure-RL look good compared to the baselines.

---

### Meta-Review · Area_Chair_Yutx · 2025-02-03

**Recommendation:** Accept (Oral)
**Confidence:** 4

**Metareview:**

This paper presents Closure-RL, a framework for discovering closures for coarse-grained discretizations of time-dependent PDEs. The scores given by reviewers are 8,7,7. The reviewers find the paper interesting, and the experiments solid. According to reviewers' comments, the proposed framework is novel, and the paper is well structured, with clear motivation and strong performance. My decision is acceptance with an oral presentation.

---

### Decision · Program_Chairs · 2025-02-11

Accept (Oral)